# The CHD Protein Kismet Restricts the Synaptic Localization of Cell Adhesion Molecules at the *Drosophila* Neuromuscular Junction

**DOI:** 10.3390/ijms25053074

**Published:** 2024-03-06

**Authors:** Ireland R. Smith, Emily L. Hendricks, Nina K. Latcheva, Daniel R. Marenda, Faith L. W. Liebl

**Affiliations:** 1Department of Biological Sciences, Southern Illinois University Edwardsville, Edwardsville, IL 62025, USA; 2Department of Biology, Drexel University, 3141 Chestnut St., Philadelphia, PA 19104, USAdmarenda@nsf.gov (D.R.M.); 3Program in Molecular and Cellular Biology and Genetics, Drexel University College of Medicine, Philadelphia, PA 19104, USA; 4Neurogenetics Program, Department of Neurology, David Geffen School of Medicine, University of California, Los Angeles, CA 90095, USA; 5Division of Biological Infrastructure, National Science Foundation, Alexandria, VA 22314, USA

**Keywords:** synapse, *Drosophila* neuromuscular junction, neuroligins, integrins, Fasciclin II, endocytosis, Endophilin B, AP-1, Rab11

## Abstract

The appropriate expression and localization of cell surface cell adhesion molecules must be tightly regulated for optimal synaptic growth and function. How neuronal plasma membrane proteins, including cell adhesion molecules, cycle between early endosomes and the plasma membrane is poorly understood. Here we show that the *Drosophila* homolog of the chromatin remodeling enzymes CHD7 and CHD8, Kismet, represses the synaptic levels of several cell adhesion molecules. Neuroligins 1 and 3 and the integrins αPS2 and βPS are increased at *kismet* mutant synapses but Kismet only directly regulates transcription of *neuroligin 2*. Kismet may therefore regulate synaptic CAMs indirectly by activating transcription of gene products that promote intracellular vesicle trafficking including *endophilin B* (*endoB*) and/or *rab11*. Knock down of EndoB in all tissues or neurons increases synaptic FasII while knock down of EndoB in *kis* mutants does not produce an additive increase in FasII. In contrast, neuronal expression of Rab11, which is deficient in *kis* mutants, leads to a further increase in synaptic FasII in *kis* mutants. These data support the hypothesis that Kis influences the synaptic localization of FasII by promoting intracellular vesicle trafficking through the early endosome.

## 1. Introduction

Synaptic function relies on the appropriate execution of processes that modify and maintain neurotransmission, endocytosis, and protein localization. These processes are regulated by cell adhesion molecules (CAMs), which physically link presynaptic neurons, postsynaptic cells, and glial cells [1]. CAMs are centrally localized at synapses where they are optimally positioned to organize neurotransmitter receptors including NMDA, AMPA, and GABA receptors [2] and active zones [3]. Thus, CAMs not only contribute to the localization of pre- and postsynaptic nanodomains but also to the apposition of functional nanodomains such that presynaptic release sites are aligned with postsynaptic receptors [4]. This alignment promotes efficient neurotransmission [5] and is disrupted in neurodevelopmental disorders [6].

Several families of CAMs contribute to the organization of synapses including, but not limited to, the immunoglobulin superfamily, neurexins and neuroligins, cadherins and catenins, and integrins. CAMs are transmembrane or glycosylphosphatidylinositol (GPI)-linked proteins that include large, glycosylated extracellular domains [7]. CAM signaling is activated by homophilic interactions between the same pre- and postsynaptic CAM extracellular domain, heterophilic interactions with other CAMs, or interactions with receptors or the extracellular matrix [8]. The intracellular domains of CAMs physically associate with scaffolding proteins, cytoskeletal regulatory proteins, and signaling pathways [9]. The specific identities and combination of synaptic CAMs and their respective downstream signaling pathways mediate synaptogenesis [10] and synaptic plasticity [11].

Synaptic CAM expression varies during development and in diseased brains. Meta-analyses of 19 transcriptomic data sets revealed that CAMs were upregulated across 17 brain regions in individuals between 0–20 years old but downregulated in individuals 20 years old or older [12]. Multiple genome-wide association studies consistently identified CAMs as strongly associated with Alzheimer’s [13] and Parkinson’s diseases [14] and in autism spectrum disorders [15]. CAM pathway genes were targets of cis-regulatory single nucleotide polymorphisms in Alzheimer’s disease patients’ temporal cortex and cerebellum compared with controls [16]. Collectively, these studies highlight the importance of CAMs for appropriate wiring and synaptic function in the brain. Despite this, little is known about the mechanisms that regulate the transcription and/or synaptic localization of CAMs.

One such mechanism of CAM transcriptional regulation is via the chromatin helicase binding domain (CHD) protein family of chromatin remodeling enzymes. Chromatin remodeling enzymes are ATPases that change the composition of histone octamer subunits within nucleosomes or the position of nucleosomes, thereby exposing or shielding access to certain DNA sequences [17]. As such, CHD proteins promote transcriptional activation of some genes and transcriptional repression of others [18]. RNA sequencing data demonstrate that genes enriched for extrinsic to membrane and focal adhesion were differentially expressed after *Chd8* knock down in human progenitor cells [19]. Transcript levels of the *neural cell adhesion molecule* (*NCAM*) were significantly reduced in *Chd7* null embryonic stem cells [20] and *Chd8* knock down human neural progenitor cells [19] but were increased in fully differentiated neurons derived from heterozygous *Chd8* loss of function neural progenitor cells [21].

Kismet (Kis) is the *Drosophila* ortholog of both *CHD7* and *CHD8*, which, when mutated, are causative for CHARGE syndrome [22] and implicated in autism spectrum disorders [23], respectively. Based on their conserved structures [24], Kis likely shares transcriptional mechanisms and targets with CHD7 and CHD8. Indeed, Kis binding is enriched in the promoters of *fasciclin II* (*fasII*), which is the *Drosophila* ortholog of *NCAM*, *neuroligins 2* and *4*, *neurexin IV*, and *dscam2* [25] in *Drosophila* intestinal stem cells. The functional consequences of these genetic perturbations in mature neurons are unknown. Therefore, we sought to better understand the synaptic mechanisms that regulate CAM expression and localization in *kis* mutants.

We used the *Drosophila* neuromuscular junction (NMJ), which is structurally and functionally similar to mammalian central nervous system glutamatergic synapses [26,27], to examine the relationship between CHD proteins and CAM expression and localization. This system is advantageous because the third instar NMJ is a mature synapse [28] that is amenable to genetic manipulation and transcriptional regulation is highly conserved between flies and humans [29]. Kis restricts synaptic levels of FasII and is important for endocytosis [30], neurotransmission, synaptic organization, and behavior in *Drosophila* larvae [31] and adults [32]. Here we show that Kis represses the synaptic localization of several CAMs, including those of the integrin and neuroligin families, but only directly regulates transcription of *neuroligin 2*. We therefore attributed the increase in synaptic CAMs as secondary to Kis activating transcription of gene products that promote intracellular vesicle trafficking including *endophilin B* (*endoB*) and/or *rab11* [30]. Knock down of EndoB in all tissues or neurons increases synaptic FasII, while knock down of EndoB in *kis* mutants does not produce an additive increase in FasII. In contrast, neuronal expression of Rab11, which is deficient in *kis* mutants, leads to a further increase in synaptic FasII in *kis* mutants. These data support the hypothesis that Kis influences the synaptic localization of FasII by promoting intracellular vesicle trafficking.

## 2. Results

Synaptic CAMs are critical for developmental processes including synaptogenesis, synapse refinement, and synaptic maintenance [33]. Aberrant expression of CAMs occurs in both neurodevelopmental disorders [15] and neurodegenerative diseases [34]. We previously demonstrated that synaptic levels of the NCAM homolog, FasII, are increased and the apposition of active zones relative to postsynaptic glutamate receptors is perturbed at *kis* mutant NMJs [31]. Given that multiple CAMs may be localized to the same synapse [33] and Kis binds to regulatory regions of *neuroligins* (*nlgs*) *2* and *4* in *Drosophila* intestinal stem cells [25], we sought to determine whether Kis may influence the expression of additional CAMs at the NMJ. We used two *kis* mutant alleles to examine CAM expression and synaptic localization including the hypomorphic allele, *kis^k13416^*, and *kis^LM27^*, a null allele [35]. Because *kis^LM27^* animals are embryonic lethal, we used larvae heterozygous for *kis^k13416^* and *kis^LM27^*.

### 2.1. kis Restricts the Synaptic Localization of CAMs

Postsynaptic Nlgs bound to presynaptic Neurexins induce synaptogenesis in neuronal and non-neuronal cells [36,37]. Nlg1 specifically recruits NMDA receptors containing GluN1, GluN2A, and GluN2B subunits in cultured hippocampal cells by interacting with their extracellular domains [38]. This domain was required for the induction of long-term potentiation in the CA1 region of the hippocampus [39]. In *Drosophila*, there is a single presynaptic homolog of Neurexin, Neurexin-1 (Nrx-1), and four Nlg homologs [26]. Overexpression of *nlg1* in postsynaptic muscles of the NMJ results in an increased number of postsynaptic boutons but reduces evoked and miniature endplate junctional currents [40] and this phenocopies *kis* mutants [31]. Therefore, we examined the synaptic localization of Nlg1 and Nlg3 in *kis* mutants. We used HRP, which recognizes neuronal N-glycans [41], to label neuronal membranes in conjunction with available antibodies for Nlgs. There was an increase in synaptic levels of both Nlg1 and Nlg3 in *kis^k13416^* mutants and of Nlg1 in *kis^LM27^*/*kis^k13416^* mutants (Figure 1A,B).

To determine if the increase in Nlg1 and Nlg3 may be attributed to transcriptional regulation by Kis, we examined the transcript levels of *nrx-1* and *nlgs* in both presynaptic neurons and postsynaptic muscle of *kis* mutants. *Nrx-1* and all four *nlgs* are expressed in presynaptic neurons, while only *nlgs1*, *-2*, and *-3* are expressed in postsynaptic muscle (Flybase.org). There was an approximate two-fold increase in *nlg2* transcripts in the CNS but not muscle of *kis^LM27^*/*kis^k13416^* mutants (Figure 1C). There were no other notable changes in *nrx-1* or *nlg* transcripts in *kis* mutants. We next examined Kis occupancy of cis-regulatory sites upstream of the *nlgs* in third instar larval central nervous systems (CNSs) via chromatin immunoprecipitation (ChIP) followed by qPCR. This was accomplished using animals expressing enhanced Green Fluorescent Protein (eGFP) within the endogenous Kis protein [42]. Kis-eGFP does not affect the localization of Kis compared with wild-type Kis [31]. We knocked down Kis in Kis-eGFP animals by expressing *UAS-kis^RNAi^* in neurons using the *elav-Gal4* driver. Knock down of Kis-eGFP using the *elav-Gal4* driver results in an approximate 55% reduction in GFP fluorescence in the ventral nerve cord [43]. Kis was enriched within both *nlg2* transcription start sites (TSS1, TSS2) in Kis-eGFP CNSs relative to Kis knock down (Figure 1D). Because our previous microarray data indicated that Kis did not affect the *dynamin* ortholog, *shibire*, mRNA levels [31], we used *shibire* as a negative control and confirmed that Kis is not enriched within the *shibire* promoter or transcription start site.

Integrins are a family of CAMs that bind to the extracellular matrix to regulate neuronal cell migration during development and synaptic maturation and function [44]. At the *Drosophila* NMJ, interactions between the Tenectin ligand and αPS2/βPS integrins promote neurotransmitter release [45]. Activity-dependent addition of NMJ boutons is restricted by interactions between postsynaptically secreted laminin A and presynaptic βν integrins [46], while growth of individual boutons and postsynaptic glutamate receptor localization is enhanced by interactions between presynaptically released Shriveled and pre- and postsynaptic βPS integrin receptors [47]. Further, knock down of *Chd7* in human neural crest-like cells led to aberrant cell migration and reduced attachment to the extracellular matrix [48], processes mediated by integrins [44]. Therefore, we also examined synaptic levels of αPS2 and βPS integrin receptors in *kis* mutants and found these integrins were increased at the NMJ of *kis^k13416^* and *kis^LM27^*/*kis^k13416^* mutants (Figure 2A,B). Similar to the Nlgs, however, *kis* mutants exhibited similar levels of *integrin receptor* transcripts in both the CNS and muscle (Figure 2C,D).

Collectively, our data indicate that Kis restricts the synaptic localization of Nlg1, Nlg3, αPS2, and βPS. Of the four Nlgs and five integrins examined, Kis only directly regulates *nlg2* transcription. These data suggest that the accumulation of CAMs at *kis* mutant synapses may be attributed to both direct transcriptional regulation by Kis and indirect regulation possibly by endocytosis and vesicle trafficking [30]. To assess these possibilities, we focused on the CAM FasII, which also accumulates at *kis* mutant NMJs [31] but is not transcriptionally regulated by Kis (Figure 2C,D, right bars). Both NCAM [49] and FasII [50,51] regulate synaptic plasticity in mature neurons.

### 2.2. Impairing Vesicle Trafficking Increases Synaptic FasII in Wild-Type Larvae but Does Not Change Synaptic FasII in kis Mutants

The synaptic vesicle cycle maintains synaptic pools in mature neurons and ensures that proteins of synaptic vesicles are appropriately sorted from those of the plasma membrane [52]. Endocytosis is part of the synaptic vesicle cycle as it enables neurons to replenish synaptic vesicles [53], maintain protein localization, and preserve the size and composition of the presynaptic membrane [54,55]. CAMs are among the cell-surface proteins internalized via endocytosis for cellular redistribution during cell migration [56]. Kis promotes endocytosis by regulating the expression of genes required for endocytosis and the localization of endocytic proteins [30]. NCAM is internalized via clathrin-mediated endocytosis [57]. Therefore, we assessed the possibility that Kis may influence FasII localization by promoting endocytosis. FasII forms homophilic interactions at the NMJ where it is expressed in presynaptic motor neurons and postsynaptic muscles [50,51]. We first examined synaptic levels of FasII after knock down of Endophilin B (EndoB), which is a BAR-domain-containing protein that facilitates membrane curvature [58]. Knock down of EndoB was accomplished by expressing *UAS-EndoB^RNAi^* in all cells using the *Actin-5c* driver, in neurons using the *elav-Gal4* driver, in postsynaptic muscle cells using the *24B-Gal4* driver, or in glial cells using the *repo-Gal4* driver. Knock down of EndoB in all cells or neurons produced an increase in synaptic FasII (Figure 3) recapitulating the *kis* mutant phenotype [31]. In contrast, knock down of EndoB in postsynaptic muscle or glia did not change synaptic FasII compared with outcrossed controls (Figure 3).

If EndoB and Kis function together in the same pathway, then simultaneous loss of function of *endoB* and *kis* should not further increase synaptic levels of FasII compared to loss of each protein individually. EndoB was knocked down in *kis^LM27^*/*kis^k13416^* mutants by expressing *UAS-EndoB^RNAi^* using the drivers listed above, except we used the *D42-Gal4* driver to knock down EndoB solely in motor neurons instead of all CNS neurons. There was no difference in synaptic FasII when EndoB was knocked down in all tissues, in motor neurons, or in postsynaptic muscle cells of *kis^LM27^*/*kis^k13416^* mutants compared with outcrossed controls (Figure 4A,B). These data support the hypothesis that Kis and EndoB work together to restrict the synaptic accumulation of FasII.

EndoB affects neuronal protein trafficking by influencing the dynamics of the endomembrane system. Although EndoB regulates endocytosis in *Drosophila* oocytes [59], it did not affect endocytosis at the NMJ. Instead, EndoB regulates autophagosome biogenesis at the NMJ [60]. EndoB facilitates synaptic vesicle recycling in *C. elegans* [61] and is concentrated on intracellular membranes instead of the plasma membrane [62], where it is implicated in endosomal trafficking via association with Rab5 and Rab7 [59]. Similarly, adaptor protein-1 (AP-1) complexes, which help form vesicles and select cargo [63], contribute to synaptic vesicle recycling [64,65] and are associated with Rab5 and the early endosome [66]. These data suggest that Kis may exhibit functional redundancy with AP-1 to restrict the localization of synaptic FasII. To assess this possibility, we knocked down the σ subunit of AP-1, which is the subunit that binds vesicular cargo [63], in *kis^LM27^*/*kis^k13416^* mutants. Similar to EndoB knock down in *kis^LM27^*/*kis^k13416^* mutants, knock down of AP-1σ in all tissues, in motor neurons, or in postsynaptic muscle cells of *kis^LM27^*/*kis^k13416^* mutants did not show an additive increase in synaptic FasII compared with outcrossed controls (Figure 4C). Taken together, these data suggest that Kis, EndoB, and AP-1σ function in the same pathway to restrict synaptic levels of FasII.

### 2.3. Increased Rab11 Activity Promotes the Synaptic Localization of FasII in Wild-Type Larvae and Shows an Additive Effect in kis Mutants

The increase in FasII at *kis* mutant synapses may be functionally linked to other proteins that help organize the synapse. Rabs are a family of GTPases that coordinate membrane trafficking between compartments of the endomembrane system [67]. *Rab11* transcript levels are reduced in *kis* mutant CNSs [30]. Rab11 specifically traffics cargo between recycling endosomes and the plasma membrane [68], thereby controlling the dynamics and concentration of membrane-associated proteins and lipids [69]. Increases in persistently active/GTP bound Rab11 also increases the number of rat cerebellar granule cell neuron terminals undergoing activity-dependent bulk endocytosis with NCAM present in bulk endosomes [70]. Thus, the loss of Rab11 at the synapse of *kis* mutants [30] may result in FasII accumulation. We assessed this possibility by expressing either constitutively active/GTP bound Rab11 (Rab11^Q70L^) or dominant negative/inactive Rab11 (Rab11^S25N^) in presynaptic motor neurons or postsynaptic muscles. While ubiquitous expression of Rab11^Q70L^ led to viable adults, ubiquitous expression of Rab11^S25N^ resulted in early larval lethality. Expression of Rab11^Q70L^ in all tissues or in postsynaptic muscle alone resulted in increased synaptic FasII compared to outcrossed controls (Figure 5A). In contrast, expression of Rab11^S25N^ in neurons but not muscles increased synaptic FasII compared to outcrossed controls (Figure 5B,C). This is contrary to *kis* mutants, which exhibit increased synaptic FasII [31] but decreased synaptic Rab11 [30].

*Kis* mutants also exhibit deficient muscle contraction and neurotransmission [31]. To assess whether constitutive Rab11 activity could restore locomotion in *kis* mutants, we expressed Rab11^Q70L^ in all neurons of *kis* mutants. *Drosophila* larval locomotion occurs by coordinated contractions of the dorsal and ventral body wall muscles [71], which are executed by a CNS central pattern generator [72]. *Kis^k13416^* heterozygous mutants expressing Rab11^Q70L^ in neurons showed an increase in both maximum speed of movement and distance travelled (Figure 6A,B) compared with the *kis^k13416^* outcrossed control. Notably, the increase in distance traveled was greater than that of the *UAS* outcrossed control indicating an augmentation of locomotion.

We reasoned that expression of constitutively active or catalytically inactive Rab11 could lead to unexpected phenotypes given that these Rab11 isoforms cannot be endogenously regulated. Further, CHD proteins affect the expression of hundreds to thousands of genes [19,20,21,25] and differential expression of secondary targets may contribute to aberrant Rab11 expression observed in *kis* mutants. Therefore, we sought to circumvent this issue by expressing the wild-type endogenous Rab11 locus under *UAS* control in *kis* mutants. If deficient Rab11 activity contributes to the accumulation of FasII at *kis* mutant synapses, then increasing Rab11 expression might restore synaptic levels of FasII. Wild-type Rab11, *UAS-Rab11^WT^*, was expressed in neurons of heterozygous *kis^LM27^* mutants. This resulted, however, in increased synaptic FasII compared to heterozygous *kis^LM27^* mutants that do not express *UAS-Rab11^WT^* (Figure 6C). Our results collectively indicate that the accumulation of FasII in *kis* mutants may be due to impaired intracellular trafficking through the early endosome but not the recycling endosome.

## 3. Discussion

The role of CHD proteins in neurodevelopment is well recognized [73] but how they function in mature neurons is poorly understood. CHD proteins affect the expression of genes and gene families required for neurodevelopment [22,23] and were identified as risk factors for neurodegenerative diseases [19,21]. Both CHD7 and CHD8 are expressed in human adult cortical neurons [74] and regulate the expression of genes involved in cell adhesion, neurotransmission, and synaptic plasticity [75,76,77,78]. We uncover an unexpected role of the *Drosophila* homolog of CHD7 and CHD8, Kis, at the synapse. Kis may facilitate the synaptic localization of the NCAM homolog, FasII, by promoting intracellular vesicle trafficking. Loss of function of *kis* or *endoB* leads to the synaptic accumulation of FasII. EndoB [59] and AP-1σ [66] assist in early endosomal transport. The increase in FasII at *kis* mutant synapses is not augmented by knock down of EndoB or AP-1σ (Figure 4). Attempting to restore Rab11 levels in *kis* mutants, however, increases synaptic FasII (Figure 6C). These data suggest that Kis, EndoB, and AP-1σ may work in the same pathway to restrict synaptic FasII, possibly by influencing FasII trafficking through the endomembrane system.

Similar to other synaptic proteins, the levels of cell surface CAMs must be tightly regulated for optimal synaptic growth. While too few CAMs in the membrane can result in retraction of the presynapse [51], too many CAMs impair the neuron’s capacity to remodel the synapse in response to changes in activity [79], negatively affecting synaptic plasticity [80]. The NCAM180 isoform is localized to synapses by diffusion from extrasynaptic sites followed by stabilization via homophilic interactions and association with the spectrin cytoskeleton [81]. CAM localization to the synaptic plasma membrane is also affected by interactions with other CAMs. N-cadherin facilitates the postsynaptic accumulation of Nlg1 in immature hippocampal neurons [82] and N-cadherin knock down in mature hippocampal neurons results in the loss of Nlg1 from the synapse [83]. Thus, it is possible that the increase in Nlg1 at *kis* mutant synapses is the result of stabilization by other CAMs as these synapses exhibit increases in Nlg1 and Nlg3 (Figure 1A,B) and the integrins σPS2 and βPS (Figure 2A,B) in addition to FasII [31]. This hypothesis presumes that the increase in *nlg2* also occurs at the synapse, is sufficient to stabilize other synaptic CAMs, and that each CAM facilitates the stabilization of every other CAM at the synapse. The latter does not occur in hippocampal neurons where the conditional knock out of β1 integrin resulted in increased localization of Nlgs to synaptosomes but of N-cadherin to lysates [84]. Thus, loss of β1 integrin increased the synaptic pool of Nlgs while decreasing N-cadherins.

Synaptic proteins are added to and retrieved from the synapse by cycling through the endomembrane system [85]. Retrieval via endocytosis occurs through a variety of mechanisms at the synapse, the best characterized of which include clathrin-mediated and activity-dependent bulk endocytosis [86]. Kis facilitates endocytosis by regulating expression of endocytic genes and localization of endocytic proteins, including the fission protein, Dynamin [30]. The deficits in endocytosis in *kis* mutants may lead to the synaptic retention of CAMs. It is also possible that the increase in CAMs at *kis* mutant synapses limits endocytosis. Indeed, membrane tension is inversely correlated with endocytosis [87] as clathrin-mediated endocytosis is slower at sites near substrate adhesion [88]. N-cadherin promotes activity-dependent endocytosis in mature cortical neurons [89]. Conversely, hippocampal *nlg1* knock out neurons exhibit an increase in activity-dependent endocytosis [90] and NCAM negatively regulates activity-dependent bulk endocytosis [91]. NCAM also promotes maturation of the endocytic machinery in cultured mouse hippocampal neurons by initially associating with the adapter protein, AP-3, and then recruiting AP-2 to the plasma membrane [3]. Thus, the relationship between synaptic levels of CAMs and endocytosis is more complex and likely influenced by the identity and localization of the CAM and developmental stage of the synapse, amongst a variety of other factors.

It is also important to consider that Kis regulates expression of both endocytic genes [30] and *nlg2*, suggesting that any relationship between synaptic levels of CAMs and endocytosis may be more complex in *kis* mutants given these and other synaptic perturbations. The indirect effects of CHD8 are amplified later in development as the number of differentially expressed genes identified in *Chd8* heterozygous [21,92] and loss of function [93] mice increases with age. Notably, *kis* mutants do not possess global perturbations in synaptic structure and organization. While Dynamin/Shibire [30] and postsynaptic glutamate receptors [31] are mislocalized in *kis* mutants relative to the active zone protein, Bruchpilot, Synapsin and Synaptotagmin, proteins required for synaptic vesicle clustering and release, are properly localized relative to Bruchpilot [30]. We also have not detected changes in actin structure or in synaptic or muscle levels of acetylated tubulin in *kis* mutants [31]. 

Our data support a model where Kis promotes early endosomal trafficking. It is unknown whether the early endosome, recycling endosome, late endosome, and lysosome are a series of compartments that arise from maturation of one compartment into another [94]. However, it is well established that endocytosed cargo are incorporated into the early endosome, which first sorts the cargo, or into endosomes that form from homotypic fusion. Rab5 serves as a marker for early endosomes, while Rab7 is a marker of late endosomes [85]. EndoB [59,62] and AP-1σ [66] are also thought to be associated with early endosomes. Kis promotes *endoB* transcription and Kis binding is enriched within the *endoB* promoter [30]. Knock down of EndoB in LNCaP cells, a human epithelial carcinoma cell line, results in increased epidermal growth factor receptor signaling due to deficient receptor endocytosis [95]. Similarly, knock down of EndoB in all tissues or presynaptic neurons results in increased synaptic FasII (Figure 3), suggesting that EndoB may also regulate intracellular trafficking in neurons. These data, coupled with our results showing that knock down of EndoB in *kis* mutants does not result in an additive increase in synaptic FasII (Figure 4A,B), indicate that Kis may regulate intracellular trafficking by transcriptional regulation of *endoB*.

AP-1 also regulates intracellular trafficking by linking clathrin to intracellular cargo in the trans-Golgi network, endosomes, and lysosomes and at the plasma membrane [96]. This central role in intracellular trafficking is responsible for the appropriate localization of apical and basolateral membrane proteins in epithelial cells [63]. Knock out of the *ap-1σ1B* subunit in mice impairs hippocampal spatial memory [65] and mutations in *ap-1σ2* in humans cause mental retardation [97]. *ap-1σ1B*^−/−^ mouse cortical neurons exhibit altered trafficking of some synaptic proteins, with some proteins mislocalized to synaptic membranes and others localized to endolysosomes [98]. These data demonstrate the importance of AP-1 protein sorting for neuronal function. Knock down of AP-1σ in *kis* mutants phenocopies knock down of EndoB in *kis* mutants by not producing an additive increase in synaptic FasII (Figure 4). The affected intracellular trafficking in *kis* mutants would also involve Rabs including Rab5 and Rab7. While we have not investigated the activity and levels of other Rabs in *kis* mutants, they may also contribute to the aberrant trafficking we hypothesize exists in *kis* mutants. Kis was enriched within *rab5* and *rab7* regulatory sites in *Drosophila* intestinal stem cells [25]. In addition, CHD8 binds to *rab5b* and *rab5b* is downregulated in *Chd8* knock down neural progenitor cells [99]. Thus, multiple lines of evidence suggest that Kis may regulate intracellular trafficking in neurons, thereby influencing the synaptic localization of CAMs including FasII, neuroligins, and integrins.

It is alternatively possible that the increase in Nlg1, Nlg3 (Figure 1), the αPS2 and βPS integrins (Figure 2), and FasII may be the result of increased translational but not transcriptional mechanisms. It is important to note that our data do not address the source of the increased CAMs in *kis* mutants. Neurons locally translate mRNAs to quickly respond to changes in activity [100]. Most rat hippocampal neuron synapses contain ribosomes in vitro and the amount of synaptic translation correlates with neuronal activity [101]. The integrin receptor mRNAs *itgb1*, *itgb2*, and *itgav* were preferentially translated in axons or dendrites compared with the soma but the *nlgn1*, *nlgn2*, *nlgn3*, and *ncam1* transcripts showed the opposite [102]. While these data would suggest that *kis* mutant synapses may exhibit less local translation due to deficient neurotransmission [31], differences in translational efficiency and protein stability also influence synaptic protein levels [103]. The first intron of the *kis* gene includes the miRNA *miR-965* [104]. Although TargetScan Fly (release 7.2) [105] does not predict that the fly transcripts for the affected CAMs in *kis* mutants contain 3′ UTR sequences recognized by *miR-965*, it is still possible that increased translation is responsible for the increase in CAMs observed at *kis* mutant synapses.

Our interpretations are also limited by the challenge of mimicking *kis* mutant phenotypes in wild-type animals. As a chromatin remodeling enzyme, Kis potentially influences the transcription of thousands of gene products [25]. Thus, as implied above, manipulating gene expression and/or expression of alternative protein isoforms in a wild-type background does not reproduce the plethora of synaptic perturbations present in *kis* mutants. CHD7 and CHD8 are implicated in neurodevelopmental disorders [22,23], which are characterized by aberrant expression of hundreds, if not thousands, of genes that collectively regulate common molecular pathways [106]. Thus, it is important to use models that emulate these conditions despite the challenges in data interpretation to gain a better understanding of the synaptic underpinnings of these conditions.

## 4. Materials and Methods

### 4.1. Drosophila Stocks and Husbandry

All fly stocks were raised and maintained in a Percival DR-36NL incubator at 25 °C with a 12 h light/dark cycle and fed Jazz Mix food (Fisher Scientific AS153). Larvae of both sexes were used for all experiments. Most fly stocks were obtained from the Bloomington *Drosophila* stock center including *w^1118^* (RRID:BDSC_3605), *kis^k13416^* (RRID:BDSC_10442), *UAS-EndoB^RNAi^* (RRID:BDSC_34935), *UAS-AP-1σ^RNAi^* (RRID:BDSC_40895), *UAS-Rab11^Q70L^* (RRID:BDSC_50783), *UAS-Rab11^S25N^* (RRID:BDSC_23261), *UAS-Rab11^WT^* (RRID:BDSC_50782), *Actin5c-Gal4* (RRID:BDSC_30558), *elav-Gal4* (RRID:BDSC_8760), *D42-Gal4* (RRID:BDSC_8816), *24B-Gal4* (RRID:BDSC_1767), and *repo-Gal4* (RRID:BDSC_7415). *Kis^LM27^* is described in [35]. *UAS-kis^RNAi.^^a^* (v109414) flies were obtained from the Vienna *Drosophila* RNAi Center.

### 4.2. Chromatin Immunoprecipitation, RNA Isolation, Reverse Transcription PCR, and qPCR

CNSs were dissected in ice cold PBS from 350–600 third instar larvae of each genotype per biological replicate. Dissected CNSs were placed in 1× PBS and stored at −80 °C. Chromatin was sheared using a Tissue Chromatin Shearing Kit with SDS Shearing Buffer (Covaris). Dissected CNSs were washed twice with 1× PBS, fixed in Buffer A with 1% methanol-free formaldehyde at room temperature for five min, and then Quenching Buffer E was applied to stop the fixation. The tissue was centrifuged at 4 °C for five min, after which the supernatant was removed. The pelleted tissue was washed twice with ice cold 1× PBS. The Wash buffer (WB) was removed and then the tissue was homogenized for 2–3 min in 500 µL Lysis Buffer B. The latter was subsequently added to increase the volume to 1 mL, followed by rocking incubation at 4 °C for 20 min 3 s of vortexing every 10 min. Lysed tissue was next pelleted, resuspended in WB C, washed, and resuspended in Covaris SDS Shearing Buffer D, which remained on the tissue for 10 min with occasional vortexing. Chromatin was sheared after transfer to a TC 12 × 12 tube for 10 min by a Covaris E220 Ultrasonicator. Sheared chromatin was visualized on an agarose gel containing 1.5% Ethidium Bromide (Fisher Scientific, Waltham, MA, USA) to confirm 100–600 bp chromatin fragments. Chromatin was then immunoprecipitated using a Magna ChIP HiSens Kit (Millipore Sigma, Burlington, MA, USA). Then, 50 µL of sheared chromatin was incubated for three hours with coated magnetic beads bound with either rabbit α-GFP (Abcam, ab290) or rabbit α-IgG (Abcam, ab171870). Chromatin was then eluted from the magnetic beads and incubated in RNase A (10 mg/mL, Fisher Scientific), for 30 min followed by incubation at 57 °C overnight in Proteinase K (10 mg/mL, Millipore). The next day, the Proteinase K was inactivated by incubating for 15 min at 75 °C. The QIAquick PCR Purification Kit (Qiagen, Germantown, MD, USA) was used to isolate DNA, which was then stored at −20 °C for qPCR.

RNA was isolated from third instar larval CNSs or muscle pelts, which were dissected from males and females. Dissections were performed in Roger’s Ringer solution (135 mM NaCl, 5 mM KCl, 4 mM MgCl_2_*6H_2_O, 1.8 mM CaCl_2_*2H_2_O, 5 mM TES, 72 mM Sucrose, 2 mM glutamate, pH 7.15). CNSs and muscle pelts were placed in nuclease-free 1.5 mL centrifuge tubes containing Invitrogen RNAlater Stabilization Solution (Fisher Scientific AM7020) and stored at −20 °C. RNA was isolated using the Invitrogen Purelink RNA Mini Kit (Fisher Scientific 12-183-025). RNA concentrations were obtained using an Implen Nanophotometer N50. Each technical replicate included 30 CNSs or eight muscle pelts per genotype. Two to three technical replicates were used to calculate relative fold changes.

QPCR Primers were designed using PerlPrimer (v. 1.1.21). RT-qPCR was performed using the iTaq Universal SYBR Green One Step Kit (Bio-Rad, 1725151, Hercules, CA, USA) and a CFX Connect Real-Time PCR Detection System (Bio-Rad). Here, 100 ng of RNA was used for each reaction. Two to three biological replicates each including three technical replicates were used for data analyses. 2^−ΔΔC(t)^ values [107] were determined by calculating the difference between the C(t) value of the target transcript reaction and the C(t) value for GAPDH to obtain ΔC(t) for each transcript. Next, the difference between the control and *kis* mutant ΔC(t)s was calculated and log transformed to obtain the 2^−ΔΔC(t)^.

### 4.3. Immunocytochemistry

Third instar larvae were fillet dissected at room temperature in Roger’s Ringer solution (135 mM NaCl, 5 mM KCl, 4 mM MgCl_2_*6H_2_O, 1.8 mM CaCl_2_*2H_2_O, 5 mM TES, 72 mM Sucrose, 2 mM glutamate, pH 7.15) on Sylgard (World Precision Instruments, Sarasota, FL, USA)-coated 60 mm dishes. The larvae were fixed for 30 min with 4% paraformaldehyde (Fisher Scientific F79500) in 1× PBS (Midwest Scientific, QS1200, Fenton, MO, USA) or in Bouin’s fixative (Fisher Scientific 112016, for FasII immunolabeling only). Fixed larvae were transferred to 1.5 mL centrifuge tubes containing PTX (1× PBS + 0.1% Triton X-100, Fisher Scientific AAA16046AP) and washed three times for 10 min each in PTX, followed by two 30 min washes in PBTX (1× PBS + 0.1% Triton X-100 + 1% Bovine Serum Albumin, Fisher Scientific BP1600-100). Primary antibodies were diluted in PBTX and applied overnight at 4 °C. Primary antibodies included guinea pig α-Nlg1 [40] used at 1:100, guinea pig α-Nlg1 (a gift from Dr. Brian Mozer) used at 1:100, mouse α-FasII (Developmental Studies Hybridoma Bank [DSHB], 1D4) used at 1:10, mouse α-βPS (DSHB CF.6G11) used at 1:50, and mouse α-αPS2 (DSHB CF.2C7) used at 1:100. After primary antibodies were removed, larvae were washed three times for 10 min each in PBTX followed by two 30 min PBTX washes. Secondary antibodies, including α-mouse FITC, α-mouse TRITC, α-rabbit FITC, and α-guinea pig FITC, were used at 1:400 and obtained from Jackson ImmunoResearch (West Grove, PA, USA). Cy3- and A647-HRP (Jackson ImmunoResearch) were applied at 1:125 with secondary antibodies. After 2 hours, PBTX washes were performed, including three times for 10 min each followed by two 30 min washes. The larvae were then placed on slides and covered with Vectashield (Vector Laboratories, H1000, Newark, CA, USA) for subsequent imaging.

Images of 6/7 NMJs within segments 3 or 4 were obtained using an Olympus FV1000 confocal microscope. Each experimental replicate used the same reagents for all genotypes. Imaging parameters for experimental replicates were determined by calculating the means of each laser intensity for control animals and applying those settings to image each experimental animal NMJ. Approximately equal numbers of controls and experimental animals were imaged each day. Image z-stacks were constructed using Fiji [108]. Mean relative fluorescence intensities were calculated from z-stacks by first obtaining the synaptic fluorescence intensity and subtracting it from the background fluorescence obtained from an area of equal size that did not include the NMJ. All experiments included at least two and up to four biological replicates, with 3–8 animals included per biological replicate. Approximately equal numbers of controls and experimental animals were used for each biological replicate. The total number of technical replicates for all immunocytochemistry experiments was greater than 11. The mean relative NMJ fluorescence intensity for each larva is represented as a point on bar graphs.

### 4.4. Larval Locomotion

Larvae were raised in standard vials. The day of the locomotion assay, larvae were transferred to a 100 mm plate containing 1.6% agar. Larvae were allowed to explore to acclimate to the crawling surface and shed debris from the home vial. After one minute, larvae were placed on a 1.6% agar arena. Locomotion was recorded for five animals at a time for 30 sec on a Canon EOS M50 camera at 29.97 frames per second. Video recordings were analyzed using the wrMTrck plugin written by Jesper S. Pedersen for Fiji. The total distance travelled in mm and maximum speed per second were calculated for 23–30 animals per experimental condition.

### 4.5. Experimental Design and Statistical Analyses

Data analyses were performed with GraphPad Prism (v. 10.1.1). Data from experiments that included a single control group were analyzed using unpaired *t*-tests. Data from experiments that included more than one control group were analyzed using a one-way ANOVA, followed by Tukey’s post hoc tests. Bartlett’s Test for homogeneity of variance was used to assess the variances between data sets. Histogram bars in figures represent the means and show sample sizes as individual points. Sample sizes indicate individual larvae, except for the RT-qPCR experiments, where the points represent one technical replicate. Statistical significance is represented on bar graphs as follows: * = <0.05, ** = <0.01, *** = <0.001 with error bars representing standard error of the mean (SEM).

## Figures and Tables

**Figure 1 ijms-25-03074-f001:**
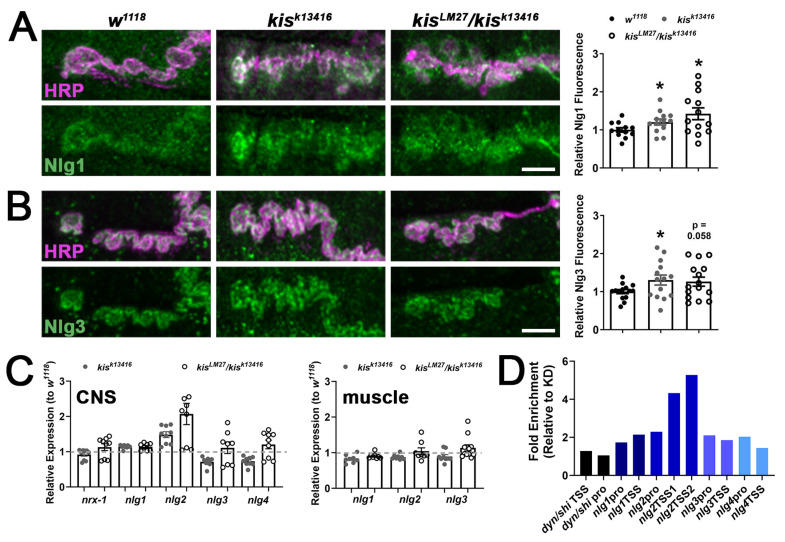
Kismet restricts the synaptic localization of Nlgs and regulates *nlg2* transcription. High-resolution confocal micrographs of 6/7 NMJ terminal boutons showing presynaptic motor neurons (magenta, HRP) and either Nlg1 (green, (**A**)) or Nlg3 (green, (**B**)). Scale bar = 5 µm. * *p* < 0.05. Right panels show quantification relative to the control, *w^1118^*. (**C**) *Nlg* transcript levels in the CNS (left) and muscle (right) of *kis* mutants. Points represent technical replicates of two or three biological replicates. (**D**) Kismet enrichment within the promoter (pro) or transcription start sites (TSS) of gene regions listed. Data shown represent two ChIP-qPCR biological replicates from CNS.

**Figure 2 ijms-25-03074-f002:**
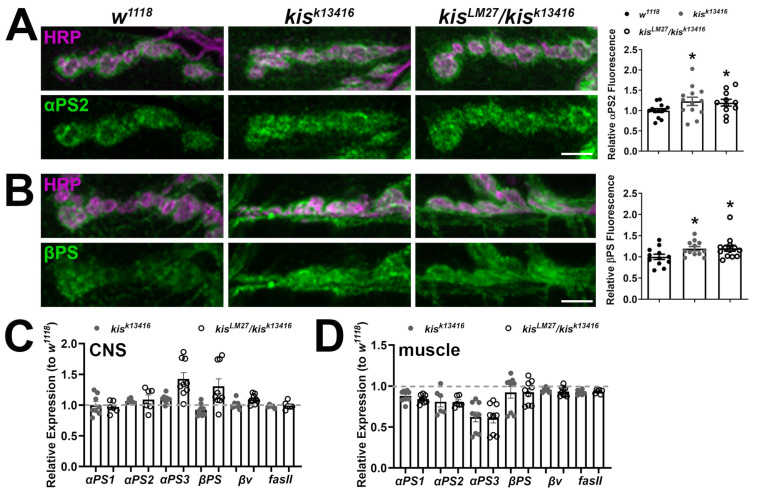
Kismet restricts the synaptic localization of αPS2 and βPS but does not influence their transcripts. High-resolution confocal micrographs of 6/7 NMJ terminal boutons showing presynaptic motor neurons (magenta, HRP) and either αPS2 (green, (**A**)) or βPS (green, (**B**)). Scale bar = 5 µm. * *p* < 0.05. Right panels show quantification relative to the control, *w^1118^*. Integrin subunit transcript levels in the CNS (**C**) or muscle (**D**) of *kis* mutants. Points represent technical replicates of two to three biological replicates.

**Figure 3 ijms-25-03074-f003:**
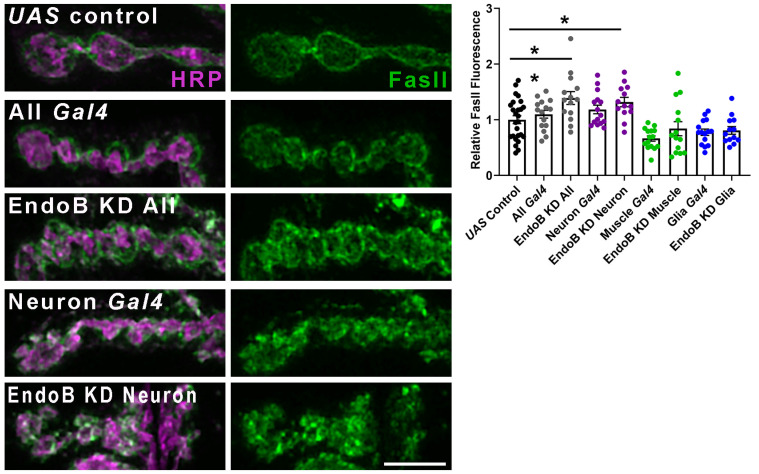
Knock down of EndoB in all tissues or neurons increases synaptic FasII. Left panels: EndoB was knocked down in all tissues (using the *Actin5c-Gal4* driver), in neurons (using the *elav-Gal4* driver), in postsynaptic muscle (using the *24B-Gal4* driver), or glial cells (using the *repo-Gal4* driver). High-resolution confocal micrographs show terminal boutons of 6/7 NMJs labeled with HRP (magenta) and FasII (green) in animals where EndoB was knocked down in all tissues or neurons. Scale bar = 5 µm. Right histogram: quantification of synaptic FasII relative to the outcrossed control, *UAS-EndoB^RNAi^/+*. * *p* < 0.05.

**Figure 4 ijms-25-03074-f004:**
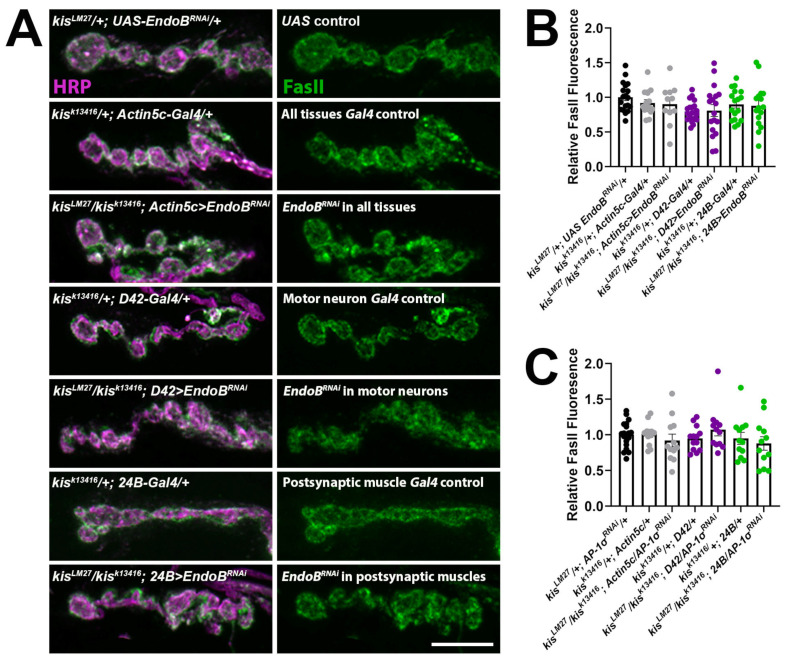
Knock down of EndoB in *kis* mutant motor neurons or muscles does not further increase synaptic FasII. (**A**) EndoB was knocked down by expressing *UAS-EndoB^RNAi^* in all tissues (using the *Actin5c-Gal4* driver), in motor neurons (using the *D42-Gal4* driver), or in postsynaptic muscle (using the *24B-Gal4* driver) of *kis* mutants. High-resolution confocal micrographs depict terminal boutons of 6/7 NMJs labeled with HRP (magenta) and FasII (green). Scale bar = 5 µm. (**B**) Histogram of synaptic FasII relative to the outcrossed control, *kis^LM27^/+*; *UAS-EndoB^RNAi^/+*. (**C**) Histogram of synaptic FasII relative to the outcrossed control, *kis^LM27^/+; UAS-AP-1σ^RNAi^/+*.

**Figure 5 ijms-25-03074-f005:**
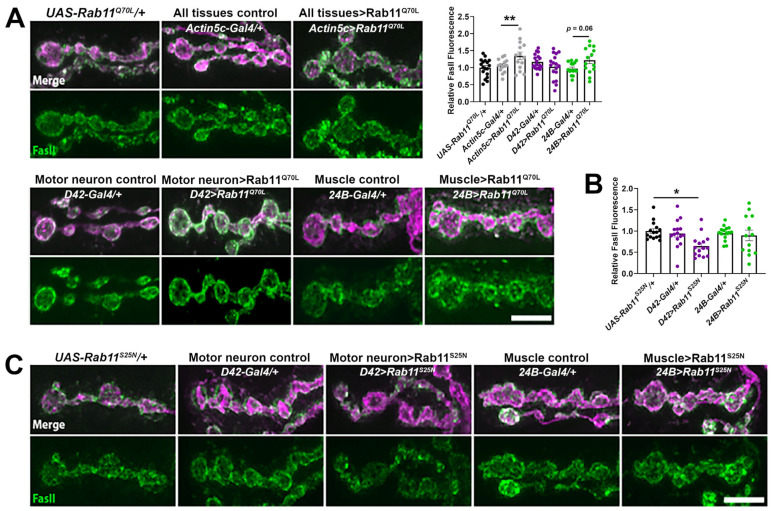
Synaptic FasII is increased when expressing a constitutively active Rab11 in all tissues but decreased when expressing a dominant negative Rab11 in motor neurons. (**A**) A constitutively active Rab11, Rab11^Q70L^, was expressed in all tissues (using the *Actin5c-Gal4* driver), in motor neurons (using the *D42-Gal4* driver), or in postsynaptic muscle (using the *24B-Gal4* driver). High-resolution confocal micrographs depict terminal boutons of 6/7 NMJs labeled with HRP (magenta) and FasII (green). Scale bar = 5 µm. Histogram of synaptic FasII relative to the outcrossed control, *UAS-Rab11^Q70L^/+*. ** *p* = 0.0046. (**B**) Histogram of synaptic FasII relative to the outcrossed control, *UAS-Rab11^S25N^/+*. * *p* = 0.011. (**C**) A dominant negative Rab11, Rab11^S25N^, was expressed in motor neurons (using the *D42-Gal4 driver*) or in postsynaptic muscle (using the *24B-Gal4* driver). High-resolution confocal micrographs depict terminal boutons of 6/7 NMJs labeled with HRP (magenta) and FasII (green). Scale bar = 5 µm.

**Figure 6 ijms-25-03074-f006:**
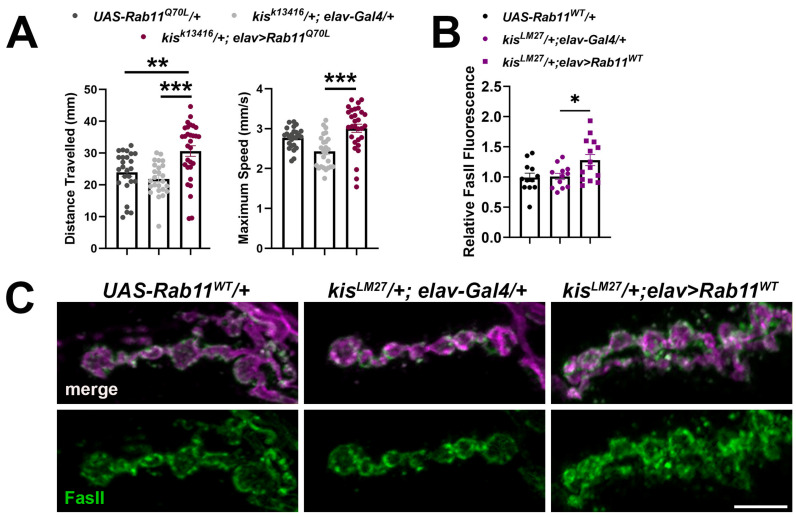
Expression of Rab11^Q70L^ and Rab11^WT^ in neurons of *kis* heterozygous mutants increases locomotion and synaptic FasII levels, respectively. (**A**) Rab11^Q70L^ was expressed in neurons of *kis^k13416^* mutants using the *elav-Gal4* driver. Histograms show larval crawling behavior on agar for 30 s quantified by wrMTrck and normalized to body lengths per second. Left histogram: ** *p* = 0.0045; *** *p* = 0.0001. Right histogram: *** *p* < 0.0001. (**B**) Quantification of synaptic FasII relative to the outcrossed control, *UAS-Rab11^WT^/+*. * *p* = 0.046. Rab11 was expressed in *kis^LM27^/+* heterozygous mutants by expressing *UAS-Rab11^eYFP^* in neurons using the elav-Gal4 driver. (**C**) High-resolution confocal micrographs of 6/7 NMJ terminal boutons showing presynaptic motor neurons (magenta, HRP) and FasII (green). Scale bar = 5 µm.

## Data Availability

The data presented in this study are available on request from the corresponding author. Data are available in the form of spreadsheets documenting data analysis of raw data.

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
