# Peer review of "The CHD Protein Kismet Restricts the Synaptic Localization of Cell Adhesion Molecules at the *Drosophila* Neuromuscular Junction"

_ijms, 2024, doi:10.3390/ijms25053074_

Round 1

Reviewer 1 Report

Comments and Suggestions for Authors

In this manuscript, Smith et al. explores the role of Kismet (Kis), the Drosophila homolog of the chromatin remodeling enzymes CHD7 and CHD8, in regulating the synaptic localization of cell adhesion molecules (CAMs). The study reveals that Kismet represses the synaptic levels of several CAMs, including Neuroligins 1 and 3 and the integrins αPS2 and βPS. While Kismet directly regulates transcription of Neuroligin 2, its influence on other CAMs is indirect, possibly through the activation of gene products promoting intracellular vesicle trafficking. Knockdown experiments suggest that Kismet, along with endophilin B and Rab11, contributes to the regulation of CAM synaptic localization through early endosome-mediated intracellular vesicle trafficking.

This research provides insights into the complex regulatory mechanisms governing the expression and localization of CAMs at synapses, highlighting the involvement of chromatin remodeling enzymes and intracellular trafficking pathways. The findings contribute to a better understanding of the molecular processes influencing synaptic growth and function in neuronal cells.

1.      In this study, additional molecular experiments, such as Western blotting, are required. These experiments can further substantiate the conclusions drawn from immunofluorescence experiments regarding changes in protein expression. Furthermore, the study has derived conclusions regarding a series of changes in protein localization, which necessitate molecular experiments to confirm that these changes are indeed alterations in localization rather than resulting from changes in protein expression.

2.      Current study acknowledges the regulatory pathways involving Kismet, Endophilin B (EndoB), AP-1σ, and Rab11 are intricate but does not fully elucidate the interactions and dependencies in these pathways. Investigate the specific molecular mechanisms underlying how Kismet (Kis) regulates the synaptic levels of cell adhesion molecules (CAMs) and its role in intracellular vesicle trafficking. Some aspects of this part are recommended: a) The study primarily focuses on specific CAMs (e.g., FasII, Nlg1, Nlg3, αPS2, βPS), and the generalizability of findings to other CAMs or synapses can be extensively explored; b) The study introduces the notion of indirect regulation of CAMs by Kismet through the activation of genes involved in vesicle trafficking, however, the specific mechanisms and potential confounders are not fully addressed; c) Investigate potential interactions between Kismet and other proteins or molecular pathways and this could include identifying proteins that act upstream or downstream of Kismet, forming a network that collectively influences synaptic regulation; d) Integrate these findings with existing knowledge in the field of neurobiology. Understanding how Kismet fits into the broader landscape of synaptic regulation can help build a more comprehensive model of the molecular processes governing neuronal function.

3.      What’s the functional consequences of altered CAM levels and vesicle trafficking? Synaptic function, neuronal connectivity, or potentially behavior in Drosophila? Understanding the physiological implications is crucial for relating these findings to broader neurobiological contexts.

Comments on the Quality of English Language

Minor editing of English language required.

Author Response

  1. We agree that Western data would allow us to determine whether total cellular levels of the cell adhesion molecules (CAMs) are increased or whether the CAMs are simply mislocalized to the synapse. Performing Western blotting experiments, however, is not possible on a tissue-specific basis at the Drosophila neuromuscular junction (NMJ). The motor neurons of the Drosophila NMJ dive under the postsynaptic muscles where branches of the motor neuron innervate sets of muscles. The innervated muscles form invaginations around motor neuron boutons. Therefore, it is not possible to separate these tissues from one another to isolate whole cells. Their nuclear contents are separable and this allows us to assess RNA levels via RT-qPCR. Although we considered performing Western blots using whole larvae, there are many additional tissues included in larvae making the interpretation of data problematic. The reviewer brings up an important point and we have added this to the description of limitations of the study in the discussion.
  2. We appreciate the suggestions to a) assess “other CAMs or other synapses”, b) identify the specific CAM vesicle trafficking mechanisms influenced by Kismet, and c) determine if Kismet is interacting with other proteins and other molecular pathways. Although we are certainly pursuing some of these research directions, each of these suggestions will require several experiments that are beyond the scope of this manuscript. If the reviewer has a suggestion for a specific experiment that will strengthen our conclusions, we would be happy to perform that experiment.
    1. We examined CAMs from three different superfamilies including the immunoglobulin (FasII), integrin, and neuroligin families. While we could extend our analyses to include the cadherin family, these data will not offer insight into the molecular pathways involved in neuronal trafficking. Investigating whether these phenotypes are present at other synapses in kismet mutants may establish the generalizability of the phenotype. These experiments could also introduce tissue-specific mechanisms that govern trafficking. Before we compare and contrast our results at the larval NMJ with other synapses, we would like to learn more about the trafficking mechanisms the reviewer describes in b.
    2. Identifying specific CAM vesicular trafficking mechanisms and how these are affected by mutations in kismet is of great interest to us. An extensive set of experiments will be required to accomplish this goal including, but not limited to, examining trafficking via compartment-specific markers in both fixed and live tissues. Given that Kismet may influence trafficking from the trans-Golgi network to endosomes and recycling endosome to the plasma membrane along with endocytosis and autophagy, these experiments will constitute another manuscript of data.
    3. We are also interested in identifying proteins that work with or oppose the actions of Kismet. Like other chromatin remodeling enzymes [1, 2], Kismet is part of a large molecular complex [3]. The identities of the components are largely unknown but are thought to differ based on developmental and tissue-specific contexts.
    4. We have done our best to integrate our findings with what is known about neuronal CAM trafficking in the discussion. If the reviewer feels that we have missed relevant literature, we are happy to integrate additional information in the discussion.
  3. We agree with the reviewer that addressing the “functional consequences of altered CAM levels and altered vesicle trafficking” is important. We describe the functional consequences in our other publications [4, 5], which are referenced in this manuscript where appropriate. Kismet mutants exhibit deficient locomotion, neurotransmission, and endocytosis. They also exhibit reductions in the synaptic vesicle pool. The latter data are included in a manuscript in review at PLoS ONE. We recognize, however, that how the perturbations introduced to kismet mutants should be functionally examined. Therefore, we have added data to Figure 6 that show that expression of a constitutively active isoform of Rab11 in kis mutants enhances their locomotion.

The reviewer also commented that minor editing of the English language was warranted. We fixed a few typos found in the manuscript but did not note systematic issues with use of articles, subject-verb plurality, etc. Note that data are plural while datum is the singular form of the noun.

  1. Hargreaves DC, Crabtree GR. ATP-dependent chromatin remodeling: genetics, genomics and mechanisms. Cell Res. 2011;21(3):396-420. Epub 20110301. doi: 10.1038/cr.2011.32. PubMed PMID: 21358755; PubMed Central PMCID: PMCPMC3110148.
  2. Meier K, Brehm A. Chromatin regulation: how complex does it get? Epigenetics. 2014;9(11):1485-95. doi: 10.4161/15592294.2014.971580. PubMed PMID: 25482055; PubMed Central PMCID: PMCPMC4622878.
  3. Dorighi KM, Tamkun JW. The trithorax group proteins Kismet and ASH1 promote H3K36 dimethylation to counteract Polycomb group repression in Drosophila. Development. 2013;140(20):4182-92. Epub 20130904. doi: 10.1242/dev.095786. PubMed PMID: 24004944; PubMed Central PMCID: PMCPMC3787758.
  4. Ghosh R, Vegesna S, Safi R, Bao H, Zhang B, Marenda DR, et al. Kismet positively regulates glutamate receptor localization and synaptic transmission at the Drosophila neuromuscular junction. PLoS One. 2014;9(11):e113494. Epub 2014/11/21. doi: 10.1371/journal.pone.0113494. PubMed PMID: 25412171; PubMed Central PMCID: PMCPMC4239079.
  5. Latcheva NK, Delaney TL, Viveiros JM, Smith RA, Bernard KM, Harsin B, et al. The CHD Protein, Kismet, is Important for the Recycling of Synaptic Vesicles during Endocytosis. Sci Rep. 2019;9(1):19368. Epub 2019/12/20. doi: 10.1038/s41598-019-55900-6. PubMed PMID: 31852969; PubMed Central PMCID: PMCPMC6920434.

Reviewer 2 Report

Comments and Suggestions for Authors

Smith et al. studied that the Drosophila homolog of the chromatin remodeling enzymes CHD7 and CHD8, Kismet, represses the synaptic levels of several cell adhesion molecules. Interestingly, their data support the hypothesis that Kis influences the synaptic localization of FasII by promoting intracellular vesicle trafficking through the early endosome.

1. It is recommended to provide the experimental material studied in the title “in Drosophila.

2. Please be advised to correct the references and bibliography according to the “Instruction for Authors.” This helps the reviewers to comprehend the text better.

3. Please provide the study's limitations and describe specifically the drawbacks.

4. Please provide the aim of the study with a specific indirect question. The current format only describes what the study performed.

5. How many animals were studied? How was the power of the study performed? How was the data distributed?

Auto-citation 3-5%. The abstract, introduction, and results are unremarkable.

Author Response

  1. We added “at the Drosophila Neuromuscular Junction” to the title.
  2. We reformatted the in text citations such that they are placed in brackets instead of superscripted. We also reformatted the bibliography such that the year of publication precedes the volume and page numbers/article identifier as suggested by the “Instruction for Authors”.
  3. We have added two paragraphs to the end of the discussion to address the study’s limitations.
  4. The aim of the study was described in the first sentence of the last paragraph of the introduction. We added another sentence to the preceding paragraph to better emphasize the aim of the study.

The number of animals for each experiment is indicated by the points in the bar graphs. We have added additional information to the methods to better describe the number of animals used for each experiment as this was a concern also raised by Reviewer #3. Statistical power is correlated with sample sizes, the expected difference in the size of results, and the p-value deemed “significant”. Given that we used 12-16 animals per condition, the results are presented as fold changes, and used the standard of p = <0.05 as statistically significant, we should achieve 80% power. Note that more animals were used for behavior because of the increased variation that is inherent in behavior. Fewer animals were used for RT-qPCR as is the standard in the literature and is described in #3 of the response to Reviewer #3. All data were normally distributed as indicated by our data analyses.

Reviewer 3 Report

Comments and Suggestions for Authors

In this study, the authors make the argument that a chromatin remodeling enzyme homolog named Kismet regulates the localization of various cell adhesion molecules (CAMs) by modulating endocytic activity in the cell. Using the larval Drosophila neuromuscular junction as a model synapse, they manipulate the expression of Kismet and several mediators of intracellular vesicle trafficking and then quantify accumulation of different CAMs at axon terminal membranes. The manuscript is well-written and control groups were chosen appropriately. Despite the use of a number of different transgenic fly lines, this is essentially a single-technique study using immunofluorescence (aside from a minor use of qPCR) from fixed-tissue samples. As such, all of the study’s claims regarding changes in endocytosis and/or vesicle trafficking are based entirely on what appear to be fairly subtle changes in fluorescence intensity of their chosen CAMs at the membrane. The “by promoting intracellular vesicle trafficking” portion of the manuscript title may be a bit of a stretch, given the lack of direct evidence of this process. Sample size also seemed somewhat low, but this was difficult to gauge based on the language used in the text. These limitations limit the potential impact of the study despite the potentially novel research questions being addressed.

Specific concerns: 1) Was blinding used for the imaging and/or analysis steps? From the methods, it sounds like the control animals for each experiment had to be known prior to imaging in order to set the imaging parameters for the whole dataset, which opens up the possibility of image selection bias.

2) I found it hard to determine the sample sizes used. Section 4.4 in the Methods states “All experiments included at least two biological replicates with approximately equal numbers of controls and experimental animals.” Yet in the histograms, most conditions showed 12 or more data points. How many data points belonged to each biological replicate? What was the upper limit of the biological replicate sample range?

3) Was any sort of statistical analysis performed for the qPCR experiments shown in Figures 1C-D and 2C-D?

Author Response

We have removed the “by promoting intracellular vesicle trafficking” from the manuscript title. We are careful in the manuscript “Kis may facilitate the synaptic localization of the NCAM homolog, FasII, by promoting intracellular vesicle trafficking” in discussion.

  1. Blinding was only used during image analyses steps as blinding cannot be achieved during confocal imaging.
  2. We have revised the methods to better describe the sample sizes. Briefly, for all immunocytochemistry and behavioral experiments, 2-4 biological replicates were used with each biological replicate including 4-8 technical replicates. We design all experiments this way to increase the inherent biological variability that exists.
  3. No, qPCR experiments most frequently do not include statistical analysis. Although some published qPCR data were statistically analyzed, most do not and this may be because of the small sample sizes. Of 10 papers published in 2022 or 2023 that used RT-qPCR to analyze expression in neurons, 8 of them used three biological replicates with three technical replicates per biological replicate.

Round 2

Reviewer 1 Report

Comments and Suggestions for Authors

Although this article still hasn't fully addressed all the questions and concerns I raised in the comments, it seems that the authors have try their best to improve it. And I realized the efforts for addressing all these suggestions are beyond the scope of this manuscript. I think this article is acceptable.